# Supratotal Resection of Glioblastoma: Better Survival Outcome than Gross Total Resection

**DOI:** 10.3390/jpm13030383

**Published:** 2023-02-22

**Authors:** Seung Hyun Baik, So Yeon Kim, Young Cheol Na, Jin Mo Cho

**Affiliations:** Department of Neurosurgery, International St Mary’s Hospital, Catholic Kwandong University, Incheon 22711, Republic of Korea

**Keywords:** glioblastoma, lobectomy, gross total resection, overall survival, supratotal resection

## Abstract

Objective: Supratotal resection (SupTR) of glioblastoma allows for a superior long-term disease control and increases overall survival. On the other hand, aggressive conventional approaches, including gross total resections (GTR), are limited by the impairment risk of adjacent eloquent areas, which may cause severe postoperative functional morbidity. This study aimed to analyze institutional cases with respect to the potential survival benefits of additional resection, including lobectomy, as a paradigm for SupTR in patients of glioblastoma. Methods: Between 2014 and 2018, 15 patients with glioblastoma underwent SupTR (GTR and additional lobectomy) at the authors’ institution. The postoperative Karnofsky performance score (KPS), progression-free survival (PFS), and overall survival (OS) were analyzed for the patients. Results: Patients with SupTR showed significantly prolonged PFS and OS. The median PFS and OS values for the entire study group were 33.5 months (95% confidence intervals (CI): 18.5–57.3 months) and 49.1 months (95% CI: 24.7–86.6 months), respectively. Multivariate analysis revealed that the O6-DNA-methylguanine methyltransferase (MGMT) promoter methylation status was the only predictor for both superior PFS (*p* = 0.03, OR 5.7, 95% CI 1.0–49.8) and OS (*p* = 0.04, OR 6.5, 95% CI 1.1–40.2). There was no significant difference between the pre- and postoperative KPS scores. Conclusions: Our results suggest that SupTR with lobectomy allows for a superior PFS and OS without negatively affecting patient performance. However, due to the small number of patients, further studies that include more patients are needed.

## 1. Introduction

Glioblastoma (GBM) is the most common malignant brain tumor. The best treatment option for GBMs is resection of as much as possible followed by chemo-radiotherapy [1,2,3,4,5,6,7]. Genetic factors that can affect the prognosis cannot be changed; the extent of resection is only one modifiable prognostic factor that may be associated with good prognosis and increased survival. A previous study reported that the median overall survival (OS) of patients with no residual tumors on postoperative magnetic resonance imaging (MRI) was significantly longer than the OS in patients with residual tumors [8].

In neurosurgery, radical resection is not considered a routine procedure, given that wide radical resection causes neurological deficits resulting from resecting normal brain tissue. In 1928, Walter Dandy reported five cases of hemispherectomy from removing gliomas in the right hemisphere [9]. After the removal of the whole right hemisphere, the patients showed no neurologic deficit apart from hemiplegia. However, the study did not show improving survival rates. One of the patients survived for 3.5 years, but all of the patients actually died soon. After this report, the benefit of removing more of the normal-appearing brain tissue was considered to not be suitable, and many neurosurgeons have since focused on removing obvious brain tumors that are visible on MRI while preserving the patient’s neurological function. However, this study had a drawback in that the number of patients was small.

Gross-total resection (GTR) is determined based on the postoperative T1-enhanced MRI. Yan Michael Li reported that the removal of lesions with a high signal intensity on T2-FLAIR MRI improved survival [8]. However, it is well known that cancer cells of a GBM directly invade into areas that appear to be normal on MRI [10,11]. Therefore, the improvement of the survival rate after removing normal brain tissue around the tumor remains unclear.

Hugues Duffau performed supratotal resection (SupTR), which involved the extensive resection of the surrounding brain tissue as well as of the lesion, which showed promising results in patients with low-grade gliomas [12]. If the GBM is located in the less eloquent area, much of the region could be resected widely. Theoretically, the additional resection of the nonfunctioning brain would cause few or no functional deficits.

We previously used to remove enhancing lesions only in GBM occurring in the frontal, temporal, or occipital lobe. We have since adopted the principle of SupTR for GBMs, and we now include a wide resection including lobectomy of GBMs. This study was conducted to review the result of GBM patients in terms of survival and patient-performance outcomes.

## 2. Methods

### 2.1. Patient Selection

This retrospective study was waived the requirement for patient informed consent, given the retrospective nature of this study. All procedures were in accordance with the ethical standards of the institutional and/or national research committee and with the 1964 Declaration of Helsinki and its later amendments or comparable ethical standards.

Between February 2014 and January 2018, 41 patients were newly diagnosed with GBM and underwent surgery at our institute involving complete tumor resection in the frontal, temporal, or occipital lobe. We excluded patients with tumors involving eloquent areas (Figure 1).

We defined SupTR as complete resection based on the finding of no residual tumors visible on the T1-enhanced postoperative MRI and an additional frontal-temporal or occipital lobectomy. MRI was taken within 48 h after surgery. Among the 41 patients, 15 underwent SupTR without invasion of the eloquent area. Patients who had a submerged occipital lobe gave up on saving vision and underwent a lobectomy. Two neuro-radiologists separately reviewed the MRI scans to ensure agreement that GTR had been achieved in each patient. The general performance status of each patient was evaluated using Karnofsky Performance Status (KPS) scores. Patients’ KPS scores were checked preoperatively and 4 weeks postoperatively. All patients received postoperative concomitant chemo-radiotherapy and adjuvant chemotherapy with temozolomide, as previously described [1]. We excluded patients with a visible residual tumor on postoperative MRI scans.

### 2.2. Lobectomy Procedure

After tumor removal, the anterior pole of the frontal lobe was removed under navigation guidance. Next, a subpial dissection was performed at the medial, lateral, and inferior surfaces of the frontal lobe, while preserving the surrounding vascular structures and olfactory nerve. The posterior margin of the frontal lobectomy was just beneath the coronal suture, which is approximately 1–2 cm anterior to the precentral sulcus. An awake craniotomy was performed in cases when the tumor was near an eloquent area. Moreover, tumor removal and/or additional anterior-temporal lobectomies were performed for cases with temporal-lobe tumors. The posterior resection margin was approximately 5–6 cm in the case of the non-dominant hemisphere and 3.5–4.5 cm in the case of the dominant hemisphere from the temporal pole. When the tumor was located in the occipital lobe, the whole occipital lobe was resected regardless of visual symptoms. Figure 2 illustrates some representative cases. We performed frontal, temporal and occipital lobectomy.

### 2.3. Molecular Diagnostics

Genetic information regarding the tumor was retrospectively collected from electronic medical records. We examined the O6-DNA-methylguanine methyltransferase (MGMT) promoter methylation status using a methylation-specific polymerase chain reaction, as previously described. Isocitrate dehydrogenase mutations were detected through direct sequencing or immunohistochemistry using the R132H mouse monoclonal antibody.

### 2.4. Statistical Analysis

The OS were analyzed by the Kaplan–Meier method, and pre- and postoperative KPS scores were compared using the Wilcoxon signed-rank test. All statistical analyses were performed using IBM SPSS software version 24.0 (IBM Corp.). A *p*-value of 0.05 was regarded as statistically significant.

## 3. Results

### Patient Characteristics

During the study period, 41 patients underwent surgery for GBM at our institution. Among them, 15 patients had GBMs confined in the frontal, temporal, or occipital lobes; further, they had undergone complete tumor resection. Table 1 summarizes the characteristics of all the patients. The median age was 64 years (range 29–79 years). Among the patients, nine (60.0%) were female and eight (53.3%) harbored a methylated MGMT promoter. The median preoperative and postoperative KPS scores were 75 (range 60–100) and 80 (range 60–100), respectively. In 14 patients, 5-ALA fluorescence guidance was used for surgery. There was no significant difference between the pre- and postoperative KPS scores (*p* = 0.356, related-samples Wilcoxon signed-rank test). The median follow-up duration was 29.8 months (95% CI 23.5–86.6 months, reverse Kaplan–Meier method). At the last observation time, 3 (20.0%) patients were dead and 12 (80.0%) were alive. The median PFS and OS values were 33.5 months (95% CI 26.5–57.3 months) and 49.1 months (95% CI 24.7–86.6 months), respectively (Figure 3).

We performed multivariate logistic progression analysis with the following variables: age, preoperative KPS score, postoperative KPS score, and MGMT methylation status. This analysis revealed that the MGMT methylation status was the only significant and independent predictor for both prolonged PFS (*p* = 0.03, OR 5.7, 95% CI 1.0–49.8) and prolonged OS (*p* = 0.04, OR 6.5, 95% CI 1.1–40.2) (Table 2).

## 4. Discussion

Many studies have reported that the extent of resection is a crucial positive prognostic factor in patients with GBM [13,14,15,16]. The extent of the resection results is positively correlated with progression-free and overall survival [15]. However, there can be tumor recurrence even after GTR and standard chemo-radiotherapy. The extent of resection was defined based on postoperative T1-weighted contrast-enhanced MRI results. Nevertheless, there can be an infiltration of GBM tumor cells well beyond contrast-enhancing areas [17,18]. Even when postoperative MRI images show the removal of enhancing lesions, there can be tumor recurrence due to infiltrating tumor cells [11]. Most recurrent tumors occur adjacent to the resection margin or within 2 cm [10,19,20].

This problem can be solved with the use of 5-ALA, which allows for a broader tumor resection, since it reveals infiltrating tumor cells beyond the enhancing region [21]. Although GTR was performed based on postoperative MRI, patients with residual 5-ALA fluorescent tumors at the surgical field showed worse outcomes than those without residual fluorescence [22]. However, in our small cases, the use of 5-ALA did not significantly affect outcomes.

Given the recent emergence of SupTR, the exact interpretation of SupTR remains unclear [23,24,25]. Duffau used this term to describe extended resection with a margin beyond the MRI-defined abnormalities in low-grade gliomas [23]. Therefore, T2 and FLAIR images were used as references for SupTR in low-grade gliomas. For GBMs, the term SupTR has been used to describe a resection beyond the contrast-enhancing lesion [26].

This study considered lobectomy as part of the SupTR procedure. This procedure resulted in a significantly larger volume of the resection cavity compared with the initial tumor volume. Therefore, a larger resection-cavity volume than the initial tumor volume can be considered as a criterion for SupTR. When it comes to resecting a tumor, definitions can vary. We think “FLAIRECTOMY” is a definition among them [18]. However, we think that our method fits the criteria. As long as the eloquent area is not invaded, resection of one lobe is considered a method of supratotal resection.

In our study, the median PFS and median OS were 33.5 and 49.1 months, while the respective values in conventional GTR are 12 and 16 months [27,28]. These findings indicated that SupTR procedures can provide clinical benefits for carefully selected patients with GBM showing non-eloquent tumor localization. This study proposes the excision of the entire lobectomy as a paradigm for SupTR. Therefore, these findings are suggestive of lobectomy as an aggressive SupTR policy that constitutes the surgical modality of choice. However, lobectomy as an oncosurgical resection tool bears the risk of a postoperative decline in language, memory, and visual loss. However, given our retrospective design, a similar analysis was beyond the feasibility of our study. Subsequent prospective study designs might allow for the assessment of neurocognitive issues and therefore provide a more comprehensive view of lobectomy as a potential seminal oncosurgical therapeutic strategy for GBM.

This study has several limitations. The main limitation was the acquisition of retrospectively collected data. Specifically, the patients were not randomized and treated according to the physician’s decision. However, to rule out the heterogeneity of non-eloquent tumor localization, we applied highly selective inclusion criteria for GBMs located in the frontal, temporal, and occipital lobes, which resulted in a small sample size. Additionally, the sample size was too small to reach a conclusion. There is a need for other large-scale studies. A comparison between the GTR group and the SupTR is considered necessary, but such a comparison was not possible due to the small number of patients in our hospital; therefore, we conducted a comparison with a conventional study. Finally, we only included data from a single center.

## 5. Conclusions

Our results suggest that SupTR with a lobectomy allows for a superior PFS and OS without negatively affecting patient performance. Our findings demonstrate that GTR plus an additional lobectomy can improve PFS and OS without functional deterioration. Therefore, GTR plus lobectomy is a valid and safe alternative to achieve supramaximal resection. The extent of resection is positively correlated with improvements in outcomes. Since the extent of resection is the only modifiable prognostic factor in patients with GBM, this surgical strategy could improve their outcomes.

## Figures and Tables

**Figure 1 jpm-13-00383-f001:**
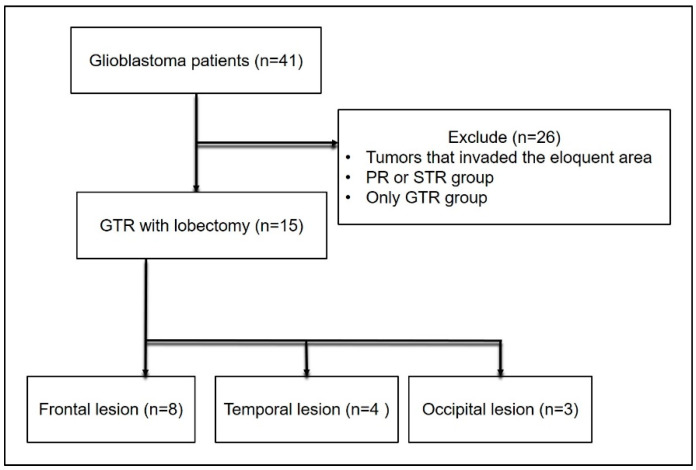
CONSORT flow diagram of disposition of patients enrolled in the study.

**Figure 2 jpm-13-00383-f002:**
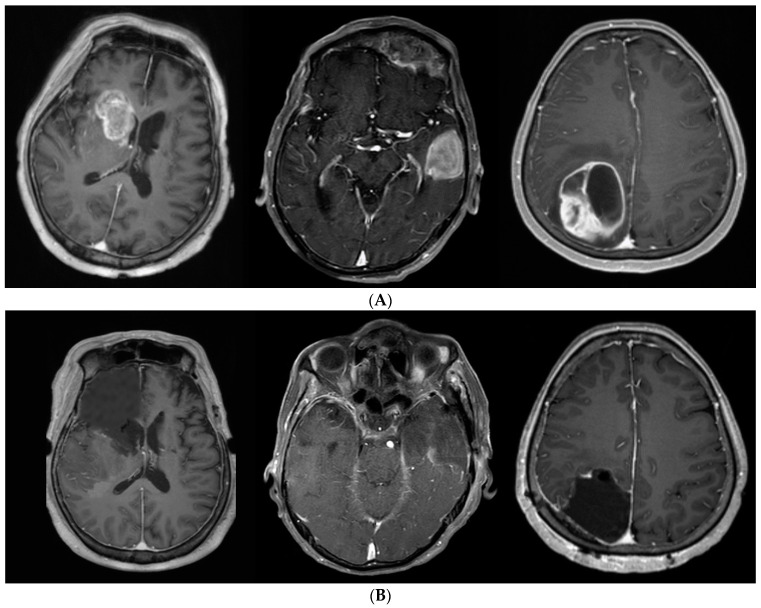
(**A**). Preoperative axial T1-weighted gadolinium-enhanced MRI showing a GBM involving the frontal, temporal and occipital lobes. (**B**). An example of a patient in whom SupTR was achieved. After complete resection of the tumor, a frontal, temporal and occipital lobectomy was performed.

**Figure 3 jpm-13-00383-f003:**
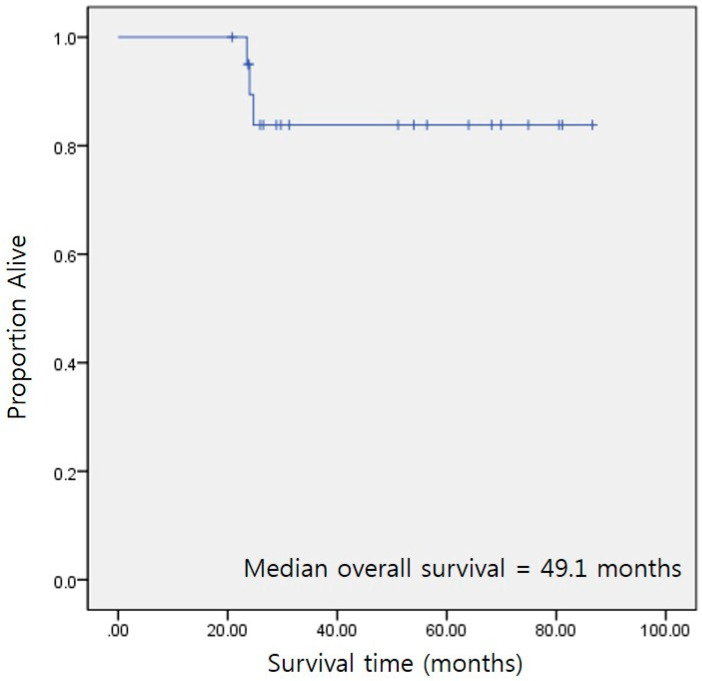
Kaplan–Meier overall survival curve in the study with a newly diagnosed GBM treated with SupTR (GTR with lobectomy).

**Table 1 jpm-13-00383-t001:** Patient characteristics.

No. of Patient	15
Median age at operation	64 (29–79)
Female sex	9 (60.0%)
KPS score pre-OP	75 (60–100)
KPS score post-OP	80 (60–100)
Frontal/Temporal/Occipital	8/4/3
5-ALA fluorescence	14 used/1 unused
IDH mutant/wild type	6 (40%)/9 (60%)
MGMT methylated/Unmethylated	8 (53.3%)/7 (46.7%)

**Table 2 jpm-13-00383-t002:** Multivariate analysis of survival.

	PFS			OS		
	Adjusted OR	95% CI	*p* Value	Adjusted OR	95% CI	*p* Value
Age	0.8	0.9–1.0	0.7	0.9	0.8–1.1	0.8
MGMT	5.7	1.0–49.8	0.03	6.5	1.1–40.2	0.04
Pre-OP KPS score	1.2	1.1–39.4	1	1.3	0.1–2.4	0.9
Post-OP KPS score	1.1	0.2–2.4	0.8	1	0.9–1.5	0.9

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
