# Peer review of "Supratotal Resection of Glioblastoma: Better Survival Outcome than Gross Total Resection"

_jpm, 2023, doi:10.3390/jpm13030383_

Round 1

Reviewer 1 Report

General comments 

The Authors presented a retrospective single center series dealing with supramaximal resection of glioblastoma. 

The topic is very interesting, the evidence in the literature related to supramaximal resection of glioblastoma is not unique, although supramaximal resection was initially identified as a topic worthy of discussion and recommendation, only a few small studies have provided Class III evidence; the aim of the authors is to try to fill the gap in the literature on this topic.

The manuscript is well-structured, however there are some English language errors with some sentences that should be revised

The aim of the study is clearly exposed, gap in knowledge is adequately addressed in introduction and discussion.

Comments for Authors 

#1

Introduction, pag 3, line 21. I think there is an error in the bibliographic citations (15 perhaps they wanted to cite 16), I recommend checking

#2

The authors used post-operative MRI to assess extent of resection. When did they perform MRI? Before 72 hours from surgery? Recent evidence and guidelines1 recommend MRI execution within 72 hours of surgery, beyond this period there may be artifacts that reduce the reliability of the exam.

#3

The Authors’ definition of supratotal is somewhat arguable. The authors defined supratotal as GTR plus frontal/temporal/occipital lobectomy. However, other Authors define the supratotal differently2; glioblastoma can circumferentially infiltrate the adjacent parenchyma  and thus further resection in only one direction cannot be considered supratotal.

Glenn3 defined supramaximal as resection of all enhancing tissue plus 1 cm of circumferential brain tissue surrounding enhancement;

Mampre4 defines supramaximal resection the complete resection of enhancing volume plus T2 hyperintense adjacent parenchyma.

Pessina5 defines supramaximal as resection of 100% of contrast enhancing volume plus all adjacent FLAIR hyperintense region (“FLAIRECTOMY”).

All these authors recommend extending the resection beyond the enhancing nodule in all directions to limit the spread of tumor cells. In the technique proposed by the Authors (see examples 1 and 2 in figure 2) the resection is extended only in an anterior direction and is not extended along the other margins of the tumor, therefore it is difficult to hypothesize that it can certainly limit tumor spread.

#4

The Authors claim to have excluded GBM in eloquent areas.

With reference to comment #3 let's consider figure 2, example 1. It is a tumor of the head of the caudate nucleus, adjacent to the internal capsule, thalamus and basal ganglia. It is a region that must be considered eloquent; the Authors carried out an excellent radical resection and added a lobectomy, the resection was correctly extended only anteriorly because in all other directions they would have damaged eloquent tissue causing deficits, therefore I believe that this case must be excluded and that it is not a true supramaximal as in the posterior, medial and posterolateral direction the resection was correctly limited to the edges of the enhancing node.

A similar argument applies to case 3, a left temporal tumor in which an extensive resection also posteriorly and superiorly could have caused aphasia.

#5

“This retrospective study was approved by Catholic kwandong Univeristy Institutional review board, which also waived the requirement for patient informed consent given the retrospective nature of this study”

This sentence is duplicated (page 4 “patients selection” and page 7 “ethical committee”).

#6

The median KPS score reported in Table 1 differs from the data reported in the Results section. I recommend checking the results carefully

#7

The Authors state that their patients have significantly higher OS and PFS, however they only compared the results with the article by Pessina et al. We believe that this method of analyzing the results is insufficient, it would be necessary to compare the results with a series of patients with similar characteristics (clinical, demographic, prognostic data…) treated with a similar protocol but only GTR without supramaximal extension. We specify that the work of Pessina (which they used for the comparison) deals with supramaximal resection is therefore not suitable for comparison.

#8

In the conclusion section the Authors affirm that  “Lobectomy significantly improves postoperative functional status”

This conclusion  is not supported by the results.

In the results the Authors showed that there was no significant difference between the pre- and postoperative KPS scores.

Similarly, the methodology of the study and the results obtained are not sufficient to state that the proposed technique allows to increase PFS and OS

Bibliography

1.        Weller M, van den Bent M, Tonn JC, et al. European Association for Neuro-Oncology (EANO) guideline on the diagnosis and treatment of adult astrocytic and oligodendroglial gliomas. Lancet Oncol. 2017;18(6):e315-e329. doi:10.1016/S1470-2045(17)30194-8

2.        Domino JS, Ormond DR, Germano IM, Sami M, Ryken TC, Olson JJ. Cytoreductive surgery in the management of newly diagnosed glioblastoma in adults: a systematic review and evidence-based clinical practice guideline update. J Neurooncol. 2020;150(2):121-142. doi:10.1007/s11060-020-03606-5

3.        Glenn CA, Baker CM, Conner AK, et al. An Examination of the Role of Supramaximal Resection of Temporal Lobe Glioblastoma Multiforme. World Neurosurg. 2018;114:e747-e755. doi:10.1016/j.wneu.2018.03.072

4.        Mampre D, Ehresman J, Pinilla-Monsalve G, et al. Extending the resection beyond the contrast-enhancement for glioblastoma: feasibility, efficacy, and outcomes. Br J Neurosurg. 2018;32(5):528-535. doi:10.1080/02688697.2018.1498450

5.        Pessina F, Navarria P, Cozzi L, et al. Maximize surgical resection beyond contrast-enhancing boundaries in newly diagnosed glioblastoma multiforme: is it useful and safe? A single institution retrospective experience. J Neurooncol. 2017;135(1):129-139. doi:10.1007/s11060-017-2559-9

Author Response

#1

Introduction, pag 3, line 21. I think there is an error in the bibliographic citations (15 perhaps they wanted to cite 16), I recommend checking

 Edited this to the text.

#2

The authors used post-operative MRI to assess extent of resection. When did they perform MRI? Before 72 hours from surgery? Recent evidence and guidelines1 recommend MRI execution within 72 hours of surgery, beyond this period there may be artifacts that reduce the reliability of the exam.

MRI was taken within 48 hours after surgery.

Edited and added this to the text.

#3

The Authors’ definition of supratotal is somewhat arguable. The authors defined supratotal as GTR plus frontal/temporal/occipital lobectomy. However, other Authors define the supratotal differently2; glioblastoma can circumferentially infiltrate the adjacent parenchyma  and thus further resection in only one direction cannot be considered supratotal.

Glenn3 defined supramaximal as resection of all enhancing tissue plus 1 cm of circumferential brain tissue surrounding enhancement;

Mampre4 defines supramaximal resection the complete resection of enhancing volume plus T2 hyperintense adjacent parenchyma.

Pessina5 defines supramaximal as resection of 100% of contrast enhancing volume plus all adjacent FLAIR hyperintense region (“FLAIRECTOMY”).

All these authors recommend extending the resection beyond the enhancing nodule in all directions to limit the spread of tumor cells. In the technique proposed by the Authors (see examples 1 and 2 in figure 2) the resection is extended only in an anterior direction and is not extended along the other margins of the tumor, therefore it is difficult to hypothesize that it can certainly limit tumor spread.

Edited and added this to the text.

“When it comes to resecting a tumor, definitions can vary. We think “FLAIRECTOMY” is a good method among them. But we think our method is one of them. As long as the eloquent area is not invaded, resection of one lobe is considered a method of supratotal resection.”

#4

The Authors claim to have excluded GBM in eloquent areas.

With reference to comment #3 let's consider figure 2, example 1. It is a tumor of the head of the caudate nucleus, adjacent to the internal capsule, thalamus and basal ganglia. It is a region that must be considered eloquent; the Authors carried out an excellent radical resection and added a lobectomy, the resection was correctly extended only anteriorly because in all other directions they would have damaged eloquent tissue causing deficits, therefore I believe that this case must be excluded and that it is not a true supramaximal as in the posterior, medial and posterolateral direction the resection was correctly limited to the edges of the enhancing node.

A similar argument applies to case 3, a left temporal tumor in which an extensive resection also posteriorly and superiorly could have caused aphasia.

In the case of invasion of the dominant hemisphere, the temporal lobectomy limit of 4.5 cm was not included by mistake. We have added this point to the text. When the basal ganglia is invaded, it is difficult to assume that all surrounding tissue has been removed. However, the significance was placed on the resection of the remaining frontal lobe, which is a non-eloquent area. We added the following sentence to the method section.

“The posterior resection margin was approximately 5–6 cm in case of non-dominant hemisphere, 3.5-4.5cm in case of dominant hemisphere from the temporal pole.”

#5

“This retrospective study was approved by Catholic kwandong Univeristy Institutional review board, which also waived the requirement for patient informed consent given the retrospective nature of this study”

This sentence is duplicated (page 4 “patients selection” and page 7 “ethical committee”).

 Delete the part on page 7.

#6

The median KPS score reported in Table 1 differs from the data reported in the Results section. I recommend checking the results carefully

 One number was entered incorrectly, so it has been corrected.

#7

The Authors state that their patients have significantly higher OS and PFS, however they only compared the results with the article by Pessina et al. We believe that this method of analyzing the results is insufficient, it would be necessary to compare the results with a series of patients with similar characteristics (clinical, demographic, prognostic data…) treated with a similar protocol but only GTR without supramaximal extension. We specify that the work of Pessina (which they used for the comparison) deals with supramaximal resection is therefore not suitable for comparison.

 I think there may be disagreements on this point. I agree with the reviewer's opinion. However, I think our thesis is meaningful because it is certain that the overall survival period has increased compared to conventional studies. Our institution has a small number of patients, so it is difficult to compare with a group that only did GTR. This part has been added to the limit part.

#8

In the conclusion section the Authors affirm that  “Lobectomy significantly improves postoperative functional status”

This conclusion  is not supported by the results.

In the results the Authors showed that there was no significant difference between the pre- and postoperative KPS scores.

Similarly, the methodology of the study and the results obtained are not sufficient to state that the proposed technique allows to increase PFS and OS

I partially agree with the reviewer's opinion. However, it seems to be correct that PFS and OS increased in our study compared to previous studies, and KPS does not seem to show a significant decrease after surgery. I think it makes sense here.

Reviewer 2 Report

The authors analyzed a retrospective cohort in order to compare the SupraTR and GTR procedure for the treatment of GBM. The study is overall in good shape, yet with some revisions can largely improve its scientific rigor. 

Major revisions:

1. In order to compare the differences of the two surgical procedures regarding to the improvement of patient survival, the researchers should present the survival date not only for the patients in the SupTR group, but also the data from the conventional GTR group. And showing the data in a Kaplan-Meier survival analysis as in Figure 1.

2. A CONSORT flow diagram should be included as the first figure of the results to indicate the inclusion and exclusion criteria.  

3. Need more explanation on the MGMT data. If MGMT methylation status is the only factor that could affect the PFS, why bother to perform SupraTR. The researchers may need to reanalyze the survival data in a 2 X 2 table like this.

Survival

SupraTR

GTR

MGMT methylation

MGMT non-methylation

Minor revisions

1. In the abstract, methods section, a more appropriate way should be 15 patients underwent SupTR, But not GTR.

2. Figure 1 illustrates the survival of the patients with SupTR, it should be presented in the results section, but not in the methods part where it describes the surgical procedure. 

3. In table 1, the contents are inconsistent with the results. Female or male, age 64 should be the median age, please specify the point, etc.

4. A detailed figure legend should be added for figure 2, including what procedure the patient received, pre-surgery and post-surgery, T1-enhanced and T2-FLAIR, etc. 

Author Response

The authors analyzed a retrospective cohort in order to compare the SupraTR and GTR procedure for the treatment of GBM. The study is overall in good shape, yet with some revisions can largely improve its scientific rigor. 

Major revisions:

1.In order to compare the differences of the two surgical procedures regarding to the improvement of patient survival, the researchers should present the survival date not only for the patients in the SupTR group, but also the data from the conventional GTR group. And showing the data in a Kaplan-Meier survival analysis as in Figure 1.

Our institution has a small number of patients, so it is difficult to compare with a group that only did GTR. This part has been added to the limit part.

  1. A CONSORT flow diagram should be included as the first figure of the results to indicate the inclusion and exclusion criteria.  

I agree with the reviewer's opinion. Edited this to the text.

  1. Need more explanation on the MGMT data. If MGMT methylation status is the only factorthat c survival data in a 2 X 2 table like this.ould affect the PFS, why bother to perform SupraTR. The researchers may need to reanalyze the

Survival

SupraTR

GTR

MGMT methylation

MGMT non-methylation

The fact that MGMT is the only factor influencing prognosis is based on intra-group comparison of supTR. When compared to the conventional study, the group with supTR has a better prognosis, and the methylated group has a better prognosis. I agree with the reviewer's opinion but our institution has a small number of patients, so it is difficult to compare with a group that only did GTR

Minor revisions

  1. In the abstract, methods section, a more appropriate way should be 15 patients underwent SupTR, But not GTR.

Edited this to the text.

  1. Figure 1 illustrates the survival of the patients with SupTR, it should be presented in the results section, but not in the methods part where it describes the surgical procedure. 

Figure 1 has been modified to Figure 2 and it is described in the result part.

  1. In table 1, the contents are inconsistent with the results. Female or male, age 64 should be the median age, please specify the point, etc.

I got it wrong by mistake because of a misspelling. I corrected it to female and marked it as median age.

  1. A detailed figure legend should be added for figure 2, including what procedure the patient received, pre-surgery and post-surgery, T1-enhanced and T2-FLAIR, etc

Edited this to the text.

Round 2

Reviewer 1 Report

General comments 

Thanks to the Authors for reviewing my comments and responses, I think the article has improved.

I believe there are other small changes that I feel are necessary before publishing the article

Comments to the Authors

#1

Abstract

I would eliminate the word "strongly"; due to the limitations of the study (perfectly analyzed by the Authors) it is not possible to state with certainty;  I think it is correct to use the phrase "Our results suggest that SupTR with lobectomy allows superior PFS and OS without negatively affecting patient performance”.

#2

Discussion:

The authors compared their results in terms of PFS and OS with the work of Pessina et al; however, as already suggested in the previous comments, it is not the article suitable for comparison, since it is an article dealing with supramaximal resection. I agree with the Authors' comment on this aspect, in their study OS and PFS are increased compared to conventional studies, I suggest using other references for comparison, the work they cite is not suitable for comparison; Authors should identify articles discussing GTR (non-supramaximal resection).

#3 

Conclusion

I suggest not to claim that “Lobectomy significantly improves functional status” because in their results functional status is not improved, there is no significant difference of KPS before and after the operation. I recommend using the same sentence used in the abstract “Our results suggest that SupTR with lobectomy allows superior PFS and OS without negatively affecting the patient performance”.

#4

I think it is excessive to state that “lobectomy might be the surgical modality of choice as an aggressive SupraTR policy for GBM”. Given the limitations of the work and the need for further clinical trials to validate the results, I would suggest a different, less strong sentence; for example GTR plus lobectomy is a valid and safe alternative to achieve supramaximal resection

Author Response

#1

Abstract

I would eliminate the word "strongly"; due to the limitations of the study (perfectly analyzed by the Authors) it is not possible to state with certainty;  I think it is correct to use the phrase "Our results suggest that SupTR with lobectomy allows superior PFS and OS without negatively affecting patient performance”.

I agree to the recommendations. I agree with reviewer's opinion and amended according to reivewer's opinion.

#2

Discussion:

The authors compared their results in terms of PFS and OS with the work of Pessina et al; however, as already suggested in the previous comments, it is not the article suitable for comparison, since it is an article dealing with supramaximal resection. I agree with the Authors' comment on this aspect, in their study OS and PFS are increased compared to conventional studies, I suggest using other references for comparison, the work they cite is not suitable for comparison; Authors should identify articles discussing GTR (non-supramaximal resection).(1) (2, 3)

 I agree to the recommendations. I agree with reviewer's opinion and amended according to reivewer's opinion.

#3 

Conclusion

I suggest not to claim that “Lobectomy significantly improves functional status” because in their results functional status is not improved, there is no significant difference of KPS before and after the operation. I recommend using the same sentence used in the abstract “Our results suggest that SupTR with lobectomy allows superior PFS and OS without negatively affecting the patient performance”.

 I agree to the recommendations. I agree with reviewer's opinion and amended according to reivewer's opinion.

#4

I think it is excessive to state that “lobectomy might be the surgical modality of choice as an aggressive SupraTR policy for GBM”. Given the limitations of the work and the need for further clinical trials to validate the results, I would suggest a different, less strong sentence; for example GTR plus lobectomy is a valid and safe alternative to achieve supramaximal resection

I agree to the recommendations. I agree with reviewer's opinion and amended according to reivewer's opinion.

Reviewer 2 Report

The author`s reply has addressed the majority of the concerns. 

Author Response

Thank you for your nice comments.